# Changes in the Content and Bioavailability of Onion Quercetin and Grape Resveratrol During In Vitro Human Digestion

**DOI:** 10.3390/foods9060694

**Published:** 2020-05-28

**Authors:** Seung Yun Lee, Seung Jae Lee, Dong Gyun Yim, Sun Jin Hur

**Affiliations:** 1Department of Animal Science and Technology, Chung-Ang University, 4726 Seodong-daero, Daedeok-myeon, Anseong-si, Gyeonggi 17546, Korea; seungyun.lee57@gmail.com; 2Immunoregulatory Materials Research Center, Korea Research Institute of Bioscience and Biotechnology 181 Ipsin-gil, Jeongeup-si, Jeonbuk 56212, Korea; seung99@kribb.re.kr; 3Department of Agricultural Biotechnology and Research, Institute of Agriculture and Life Sciences, Seoul National University, Seoul 08826, Korea; tousa0994@naver.com

**Keywords:** in vitro human digestion system, onion quercetin, grape resveratrol, free radical scavenging activity

## Abstract

We investigated the effects of in vitro human digestion on the content and bioavailability of onion quercetin and grape resveratrol caused by the composition of saliva or gastric, duodenal, or bile juice. We observed the digestibility of extracted onion quercetin and grape resveratrol, respectively, in the small intestine of the in vitro human digestion system. By liquid chromatography–mass spectroscopy, we found that the degradation of quercetin and resveratrol was influenced by small intestine digestion. Before and after in vitro human digestion, the 2,2-diphenyl-1-picrylhydrazyl (DPPH) radical scavenging activities of homogenized water- and ethanol-extracted grapes were higher than those of onion extracts. DPPH radical scavenging activity in both quercetin and resveratrol was decreased by in vitro digestion. These results will improve our understanding of how human digestion influences the contents and free radical scavenging activities of quercetin and resveratrol.

## 1. Introduction 

Quercetin compounds are natural products that are the major dietary flavonoids in onion skins [1]. Generally, quercetin consists of two aromatic rings linked by an oxygen-containing heterocycle. Quercetin’s molecular structure may play crucial roles in its extensive and well-documented biological activities. Quercetin has various health benefits, including antioxidant, anti-inflammatory, anti-obesity, anti-bacterial, and anti-cancer properties [2,3,4,5,6]. Onion quercetin and its derivatives increase the enzymatic activity of the intestinal microbiota, the antioxidant activity of blood and antiplatelets [7,8]. In addition, quercetin was developed to improve stability and solubility using an encapsulation system for the development of functional foods [9].

Resveratrol is a natural polyphenol and can be extracted or isolated from grape skins and the by-products of wine production [10]. Resveratrol has widely demonstrated biological and pharmacological activities such as a protective effects in liver disease [11], ethanol toxicity [12], high-fat diet [13], anti-cancer properties [14], anti-inflammatory effects [15] and ameliorative effects on hypertension, atherosclerosis and thrombosis [16]. In addition, many molecular targets of resveratrol such as kinases, transcription factor, cytokines, and enzymes have been correlated to clinical conditions in human [17]. Recently, the beneficial effects of resveratrol have been shown in both in vitro and in vivo models [18,19] and in clinical studies of various diseases [9,20]. Therefore, there is considerable interest in using resveratrol and quercetin as the most representative antioxidant materials in functional foods.

In vitro human digestion systems are widely used to study the structural changes, digestibility, and release of food components under simulated gastrointestinal conditions [21]. Although a number of studies have reported that the phytochemicals quercetin and resveratrol have various bioavailabilities, the influence of in vitro human digestion on the bioavailability of quercetin and resveratrol remains to be elucidated. Therefore, the purpose of this study was to determine the effects of in vitro human digestion on the contents and bioavailability of onion quercetin and grape resveratrol.

## 2. Materials and Methods

### 2.1. Materials 

Onion and grapes were obtained from a local market (Seoul, Korea). Quercetin, resveratrol, DPPH, butylated hydroxytoluene (BHT), α-amylase (~50 U/mg), pepsin (≥250 units/mg), pancreatin (4 × USP/g), lipase (100–500 units/mg), uric acid, mucin, bovine serum albumin, bile and acetonitrile were purchased from Sigma Chemical Co. (St. Louis, MO, USA). All other reagents were of the highest grade commercially available.

### 2.2. Preparation of Sample and Extracts

The onion and grapes (including skin) were washed under running tap water before being chopped into pieces. Then, they were oven-dried at 45 °C for 2 days and ground to a powder. The samples were extracted with water and 95% ethanol. The 95% ethanol extracts were prepared three times with 95% ethanol and filtered with Whatman #1 filter paper at room temperature (25 °C). The filtrate was dried using an evaporator (EYELA, Tokyo, Japan) at 40 °C. Powder samples were suspended and extracted with 1 L of water at 75 °C for 3 h. Extracts were produced using the same aforementioned filter paper and then by centrifuging at 11,200× *g* for 30 min. In addition, crude liquid was homogenized with the same volume of water concentrated using a rotary evaporator, and then lyophilized for 3 days. The freeze-dried powder sample was stored at −30 °C until use.

### 2.3. In Vitro Digestion System and Digestibility in Small Intestine

The in vitro human digestion system was constructed according to a previously described method [21]. During the in vitro human digestion, these samples were swirled at 120 rpm in a shaking water bath (Model HB-205SW, Hanbaek, Co., Korea) to simulate gastrointestinal tract motility (Table 1). In addition, the digestibility of quercetin and resveratrol during in vitro human digestion was determined by the infiltration rate of quercetin and resveratrol through dialysis tubing, determined by quantifying the amounts inside and outside the dialysis bags using a previous method [21,22]. A dialysis tubing method was used to simulate the small intestine epithelium structure and small intestine digestion. Briefly, the dialysis tubing was purchased from Spectrum Laboratories Inc. (Rancho Dominguez, CA, USA) with a molecular weight cut-off of 12–14 kDa. The dialysis tubing was cut into 10 cm pieces and soaked in distilled water at 4 °C before use. Onions and grapes were mixed with 6 mL of simulated saliva fluid (pH 6.8) and stirred for 5 min at 37 °C to mimic oral digestion. Next, to simulate stomach digestion, 12 mL of simulated gastric fluid (pH 2) was added and the mixture was stirred for 2 h at 37 °C. After stomach digestion, a sodium phosphate buffer (pH 6.9–7.0) was transferred to a dialysis tube, and the prepared dialysis tube was placed into the above-mentioned digested stomach. Next, small intestine digestion was mimicked by adding 12 mL of duodenal juice and 6 mL of bile juice, and the mixture was stirred for 2 h at 37 °C. The digestibility percentage was calculated as follows: {sample concentration inside the dialysis tube/(sample concentration outside the dialysis tube + sample concentration inside the dialysis tube)} × 100.

### 2.4. Measurement of Quercetin and Resveratrol by High-Performance Liquid Chromatography (HPLC)

The amount of onion quercetin and grape resveratrol were analyzed using HPLC (HP Agilent 1100, Hewlett Packard Co, CA, USA). The protocol was modified from previous studies (Hur et al., 2015); a Fortis H_2_O column (250 mm × 4.6 mm, 3 μm) with a linear gradient of solution A (acetonitrile, 0–70%) and solution B (water containing 0.1% formic acid) at a flow rate of 1.5 mL/min were determined. The volume of the sample injected for analysis was 20 μL, and the detection wavelength was set at 370 nm for quercetin; the compartment sample resulting from the in vitro digestion step (mouth, stomach and small intestine) including quercetin in onion. For resveratrol, the mobile phase was determined with a linear gradient of solution A (acetonitrile, 0–70%) and solution B (water containing 0.2% formic acid). The UV detector wavelength was set at 306 nm, the flow rate was 0.8 mL/min, and the injection volume was 20 μL. All solutions were passed through a 0.45 μm Whatman membrane filter before injection onto the HPLC column.

### 2.5. Liquid Chromatography-Mass Spectroscopy (LC-MS) Analysis

Electrospray ionization mass spectrometry (ESI/MS) was performed with an Agilent 1100 Series LC/MSD VL model in positive scan mode for mass analysis and detection. Following the optimization of the settings, negative ion mass spectra of the column eluate were recorded in a range of 10–1500 *m/z*. The instrument was operated with an electron multiplier ion source, and the injection volume was 20 μL. The quantifications of quercetin and resveratrol were analyzed with a comparison with the main peak of the m/z of these standards. 

### 2.6. DPPH Radical Scavenging Activity 

The DPPH radical scavenging activity was measured using a colorimetric method [23]. In total, 0.4 mL of samples was reacted with 0.4 mL of DPPH solution in methanol. The reaction mixture was incubated in the dark for 25 min. The DPPH scavenging activity was measured at a wavelength of 517 nm. Controls were prepared in the same manner using water or ethanol instead of a sample. The DPPH radical scavenging activity (%) was determined as follows:DPPH radical scavenging activity (%) = (1 − (absorbance of sample/absorbance of control)) × 100 

### 2.7. Statistical Analysis

Experiments were performed in triplicate, and the resulting data were statistically analyzed. The data were expressed as the mean ± standard deviation (SD). Statistical analyses were completed using the Tukey’s multiple comparison test. Graph Pad Prism software version 4.00 (Graph Pad Software Inc., San Diego, CA, USA) was used.

## 3. Results and Discussion

### 3.1. Identification of Quercetin and Resveratrol 

The concentrations of quercetin and resveratrol, respectively, were identified using the HPLC and LC-MS methods. The analysis of quercetin is shown in Figure 1, the chromatograms are shown in Figure 1A,C,E,G and the mass spectrums are shown in Figure 1B,D,F,H. The amount of quercetin decreased after small intestine digestion; however, the mass spectrum of quercetin after each in vitro human digestion step did not change based on a quercetin main peak with m/z 64.1 (M)^+^ on LC/MS spectra. This result indicates that the quantification of quercetin is less influenced by in vitro human digestion than the quantification of resveratrol. The results in this study are not totally in line with Boyer et al., (2005), who reported that pure quercetin and quercetin-3-glucoside were also more stable at a lower pH than higher pH, resulting in recovery changes by intestinal digestion. The detected resveratrol quantities are shown in Figure 2, the chromatograms are shown in Figure 2A,C,E,G, and the mass spectra are shown in Figure 2B,D,F,H. As shown in Figure 2, we confirmed that the resveratrol did not change until stomach digestion; however, the mass spectrum of resveratrol decreased after small intestinal digestion based on a resveratrol main peak with m/z 102.2 (M)^+^ on LC/MS spectra. Basically, the differences in the concentration of resveratrol can result from the activity of digestive enzymes that can cause the release of resveratrol in the stages from stomach to small intestine [24]. Moreover, significant changes were observed in amounts of resveratrol, characterized by the maximum increase in the contents in the small intestine and a significant reduction in the large intestine. These differences can result from a gradual release and increase in the contents of resveratrol at the stages from the stomach to the small intestine as a result of the action of digestive enzymes on the raw material; then, a reduction of concentration is caused by microflora of the large intestine [24]. This result indicates that the quantification of grape resveratrol was influenced in small intestine digestion.

In the model of digestion used in this study, the pH shifts dramatically from the stomach to the small intestine, from pH 1.5 to pH 7.5, mainly because bile salts result in a higher pH. This change in pH is the primary factor involved in the irreversible breakdown of quercetin or resveratrol; moreover, their hydrogen bonds can be cleaved by hydrolytic enzymes such as amylase, pancreatin, or pepsin. The polyphenols of flavonoid groups are known to be affected by temperature, pH, and various enzymes. In this study, the bioavailability (DPPH radical scavenging activity) of quercetin and resveratrol was found to be influenced by changes or the destruction of their contents, as they are liberated from the strong bonds of phenolic compounds with surrounding tissues through hydrogen and carbon bonding from hydroxyl or phenyl groups by temperature, ionic strength, pH, or digestive enzymes during in vitro human digestion. Therefore, this result could be presumed to show that the degradation of hydroxyl or phenyl groups in quercetin and resveratrol during in vitro human digestion may be influenced by changes in pH or hydrolysis enzymes. 

### 3.2. Changes in Contents and Digestibility of Onion Quercetin and Grape Resveratrol during In Vitro Human Digestion

The changes of the onion quercetin and grape resveratrol concentrations during in vitro human digestion are shown in Table 2. The amounts of onion quercetin in homogenized water and ethanol extract samples were reduced from 5.23 to 3.90 mg, 5.22 to 4.18 mg, and 5.99 to 5.33 mg, respectively. Basically, quercetin has low water solubility (0.01 mg/mL, 25 °C) and moderate ethanol solubility (4.0 mg/mL, 37 °C) [25,26]. In addition, a previous study also reported that quercetin and quercetin 3-glucoside levels in commercial chartreuse onion were 127.92 and 24.16 mg/g, respectively [27]. However, this study analyzed total quercetin contents without a distinction between free quercetin and quercetin glycoside. Grape resveratrol levels in homogenized water and ethanol extract samples were reduced from 1.56 to 0.89 mg, 1.15 to 1.09 mg, and 1.24 to 1.09 mg after in vitro human digestion, respectively.

The digestibility of the onion quercetin and grape resveratrol during in vitro human digestion is shown in Table 3, as determined (in the small intestine step only) by the infiltration rates of quercetin and resveratrol through dialysis tubing and by quantifying the concentration present inside and outside the dialysis bags. The digestibility of onion quercetin after homogenization and water and ethanol extraction was 45.68%, 40.14%, and 46.48%, respectively, while the grape resveratrol level was 52.45%, 51.08% and 49.54%, respectively. This result revealed that the digestibility was similar for onion quercetin and grape resveratrol during in vitro human digestion (i.e., not significantly different).

Basically, quercetin derivatives could influence their absorption rate in the small intestine and stomach during in vivo human digestion [28,29]. In our study, the change in quercetin observed during in vitro small intestine digestion may be due to the difference in pH or enzymes between the stomach and small intestine [30]. In addition, we found that the quercetin concentration in buckwheat extracts and onion extract was decreased by in vitro human digestion, and the antioxidant activity was increased by in vitro human digestion [30,31]. This increase is because quercetin aglycone was cleaved from rutin glucoside in onion [31]. Indeed, quercetin aglycone has strong antioxidant activity compared to rutin glucoside. Moreover, quercetin recovery from the onion was lower than the quercetin single compound [32].

Resveratrol in a trans-form is stable in acidic conditions, whereas its degradation begins to increase exponentially above pH 6.8 in the UV/VIS spectra using HPLC [33]. Sessa et al. (2011) also found that resveratrol remained stable, with no significant alteration in the quality and quantity of the encapsulated resveratrol during in vitro digestion [34]. Resveratrol instability is due to the deprotonation of the molecule being impacted by basic media with subsequent auto-oxidation and polymerization or degradative processes [35]. In contrast, the stability of resveratrol in liquid form can be enhanced by decreasing the temperature and pH and restricting the exposure to oxygen and light [33].

Most dietary polyphenols appear to be quite stable under stomach digestion as no significant alterations have been found in the phenolic acid and the flavonoid concentration of foods under stomach conditions [36]. Our previous study found that polyphenols are stable during gastric digestion [30]. These polyphenols are very sensitive to the mild alkaline conditions in the small intestine; some compounds may be formed of other components and structures in the duodenum [36]. As of these results, we suggest that pH, enzymes, temperature, oxygen, or carbon may influence the phenolic compositions.

### 3.3. DPPH Radical Scavenging Activity during In Vitro Human Digestion

The homogenized water and ethanol extracted samples of onion and grapes were shown to have high scavenging activities of DPPH radicals during in vitro human digestion in this study. As shown in Figure 3, grapes in particular had higher DPPH radical scavenging activities compared to onion (Figure 3). In onion, the DPPH radical scavenging activities during in vitro human digestion were 58.43%, 61.67%, 62.55%, and 63.58%. During in vitro human digestion, water-extracted samples exhibited 58.43%, 59.83%, 60.94%, and 61.95% scavenging activities, and ethanol-extracted samples exhibited 57.55%, 61.79%, 62.42%, and 63.18% DPPH radical scavenging activity (Figure 3A). During in vitro human digestion, the DPPH radical scavenging activities of homogenized grape were 72.96%, 74.32%, 75.20%, and 75.81%. The DPPH radical scavenging activities of water-extracted grapes were 66.60%, 67.12%, 67.49%, and 68.21%, and ethanol extract showed 67.36%, 68.26%, 68.76%, and 69.60% scavenging activity. However, the DPPH radical scavenging activities were not significantly different among the homogenized water and ethanol-extracted samples (Figure 3B). The DPPH radical scavenging activities of the quercetin standards were significantly reduced by in vitro human digestion (from 58.43% to 9.33%; Figure 3C), whereas the DPPH radical scavenging activities of standard resveratrol were slightly reduced by in vitro human digestion (from 72.96% to 55.99%; Figure 3D).

Numerous studies have reported that quercetin and resveratrol have radical scavenging activity [31,37]. In vitro human digestion increases the antioxidant activity of onion extracts, and the increased quercetin isolation from onion extract by in vitro human digestion is due to the increase in the antioxidant activity of onion extract [30]. Furthermore, phenolic compounds in berries exhibit enhanced antioxidant activities following in vitro digestion, resulting in the confirmation of their stability [38,39]. The antioxidant activity of flavonoids—e.g., quercetin and resveratrol—depends on the presence and number of the free hydroxyl groups in their skeleton. Therefore, the change in antioxidant activity may be closely related to the degradation of quercetin and resveratrol by in vitro digestion in this study. As a result, the DPPH radical scavenging activity of the quercetin and resveratrol standards were reduced by in vitro human digestion. The quercetin standard was more sensitive than the resveratrol standard to in vitro human digestion. In contrast, resveratrol was shown to more be stable than quercetin during in vitro human digestion in this study. This result may be attributed to the difference in the purity of quercetin and resveratrol, and other ingredients such as water; other phenolic compounds in onions and grapes could also have interfered with the results. Moreover, the results in this study are consistent with Boyer et al. (2005), who reported that quercetin aglycone may be more susceptible to oxidation or degradation under the pH conditions and digestive enzymes in the stomach and intestine. Thus, the difference between quercetin and resveratrol in DPPH radical scavenging activity may be related to the impact of aglycone which is made unstable by pH and digestive enzymes.

## 4. Conclusions

This study was a primary investigation to analyze the relationship between the contents and free radical scavenging activities of onion quercetin and grape resveratrol during in vitro human digestion. We found that in vitro human digestion influenced the contents of onion quercetin and grape resveratrol, and free radical scavenging activity was decreased by in vitro human digestion. Grape resveratrol was more stable than onion quercetin during in vitro human digestion; furthermore, the quercetin standard was more stable than the resveratrol standard. These results will improve our understanding of how human digestion influences the contents and bioavailability of quercetin and resveratrol. Although this study found that in vitro human digestion affects the content and bioavailability of onion and grape-derived natural compounds, there is clearly an urgent need for more research into the correlations between the structural changes or the bioavailability and influencing factors of the human digestion of onion and grape.

## Figures and Tables

**Figure 1 foods-09-00694-f001:**
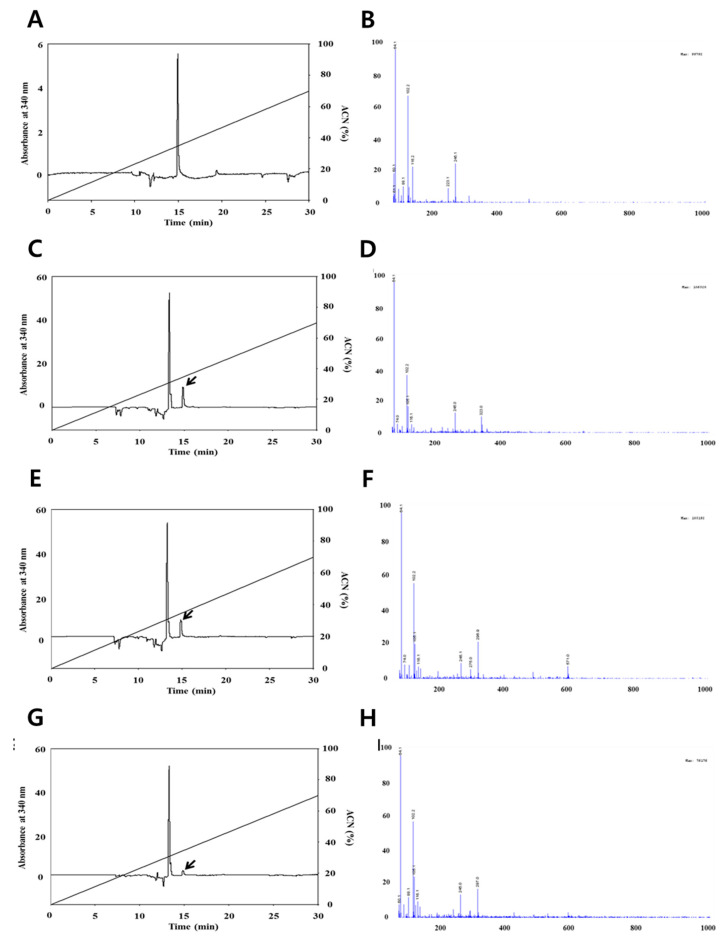
High-performance liquid chromatography (HPLC) chromatograms (left figure) and liquid chromatography–mass spectroscopy (LC/MS) spectra (right figure) of quercetin by the in vitro digestion system. ACN; Acetonitrile. (**A**,**B**) Standard quercetin, (**C**,**D**) mouth digestion, (**E**,**F**) stomach digestion, (**G**,**H**) intestine digestion. HPLC trace on a Fortis H_2_O column (250 mm × 4.6 mm, 3 μm) of the quercetin peak from standard. All HPLC operations were carried out with a linear acetonitrile gradient (0–70%) and 0.1% formic acid in water, and at a flow rate of 1.5 mL/min using a UV detector at 370 nm.

**Figure 2 foods-09-00694-f002:**
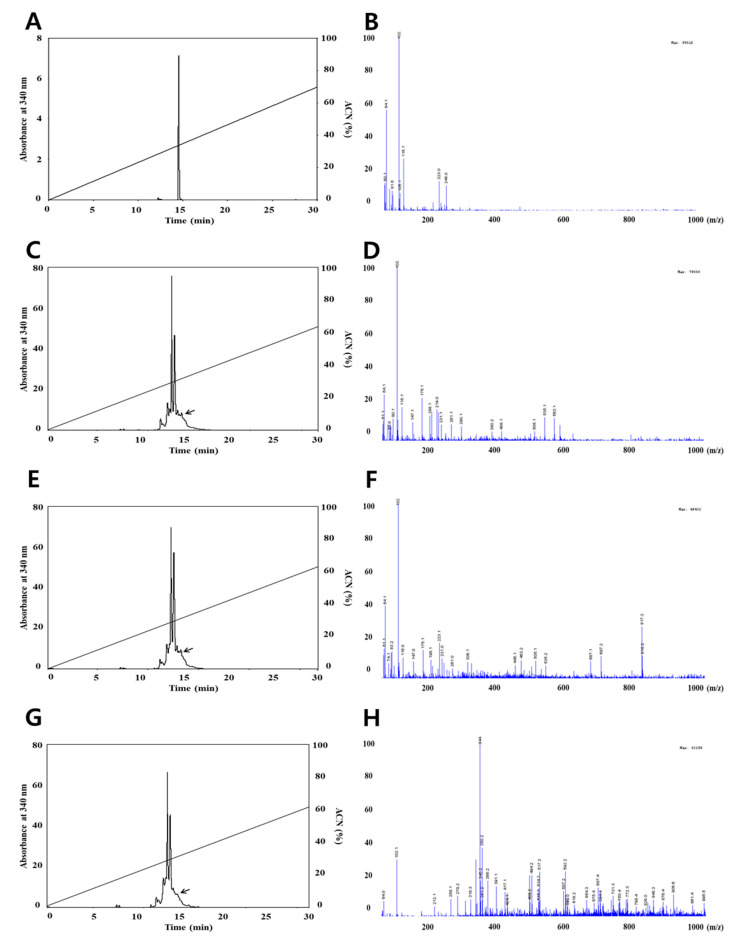
HPLC chromatograms (left figure) and LC/MS spectrum (right figure) of resveratrol by in vitro digestion system. ACN; Acetonitrile. (**A**,**B**) Standard resveratrol, (**C**,**D**) mouth digestion, (**E**,**F**) stomach digestion, (**G**,**H**) intestine digestion. HPLC trace on a Fortis H_2_O column (250 mm × 4.6 mm, 3 μm) of the resveratrol peak from standard. All HPLC operations were carried out with a linear acetonitrile gradient (0–70%) and 0.2% formic acid in water, and at a flow rate of 0.8 mL/min using a UV detector at 306 nm.

**Figure 3 foods-09-00694-f003:**
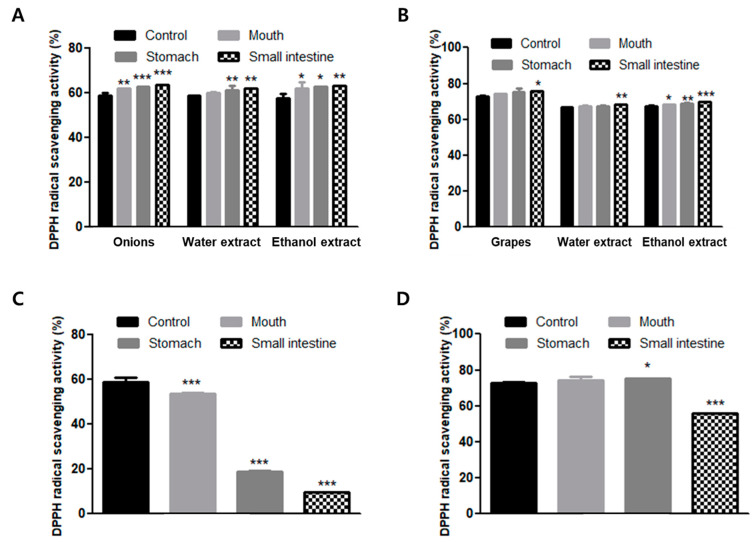
Changes in 2,2-diphenyl-1-picrylhydrazyl (DPPH) radical scavenging activities of (**A**) onion, (**B**) grapes, (**C**) quercetin and (**D**) resveratrol in the in vitro digestion system. Values are expressed as the mean ± SD of determinations made in triplicate experiments. * *p* < 0.05, ** *p* < 0.01, *** *p* < 0.001 compared with the control for each sample.

**Table 1 foods-09-00694-t001:** Components of in vitro digestion condition.

	Saliva	Gastric Juice	Duodenal Juice	Bile Juice
Organic and inorganic components	1.7 mL NaCl ^a^ (175.3 g/L) ^b^	6.5 mL HCl (37 g/L)	6.3 mL KCl (89.6 g/L)	68.3 mL NaHCO_3_ (84.7 g/L)
	8 mL urea (25 g/L)	18 mL CaCl_2_·2H_2_O (22.2 g/L)	9 mL CaCl_2_·2H_2_O (22.2 g/L)	10 mL CaCl_2_·2H_2_O (22.2 g/L)
	15 mg uric acid	1 g bovine serum albumin	1 g bovine serum albumin	1.8 g bovine serum albumin
				30 g bile
Enzymes	290 mg α-amylase	2.5 g pepsin	9 g pancreatin	
	25 mg mucin	3 g mucin	1.5 g lipase	
pH	6.8 ± 0.2	1.50 ± 0.02	8.0 ± 0.2	7.0 ± 0.2

^a^ The numbers are the concentration of chemicals to make digestive juices. ^b^ The number in parentheses are the concentration of inorganic or organic components per liter distilled water. After mixing all ingredients (inorganic components, organic components and enzymes), the volume was increased to 500 mL with distilled water.

**Table 2 foods-09-00694-t002:** Change in contents of the extracted in onion quercetin and grape resveratrol by different methods in the in vitro digestion system.

		Quercetin (mg)/Onion (g)	Resveratrol (mg/)/Grape (g)
Homogenized	Before digestion	5.23 ± 0.18 ^a^	1.87 ± 0.64 ^a^
	Mouth	5.05 ± 0.22 ^a^	1.56 ± 0.51 ^a^
	Stomach	4.77 ± 0.11 ^a^	1.54 ± 0.86 ^a^
	Small intestine	3.90 ± 0.26 ^b^	0.89 ± 0.47 ^a^
Water extract	Before digestion	5.22 ± 0.21 ^a^	1.47 ± 0.26 ^a^
	Mouth	4.41 ± 0.30 ^ab^	1.40 ± 0.47 ^a^
	Stomach	4.37 ± 0.44 ^b^	1.15 ± 0.54 ^a^
	Small intestine	4.18 ± 0.27 ^b^	1.09 ± 0.65 ^a^
Ethanol extract	Before digestion	6.02 ± 0.28 ^a^	1.97 ± 0.71 ^a^
	Mouth	5.99 ± 0.17 ^a^	1.66 ± 0.65 ^a^
	Stomach	5.82 ± 0.29 ^ab^	1.24 ± 0.48 ^a^
	Small intestine	5.33 ± 0.09 ^b^	1.09 ± 0.64 ^a^

Data are represented as the mean ± standard deviation. *n* = 3. ^a,b^ means with different superscript letters in a column within each digestion step differ significantly (*p* < 0.05).

**Table 3 foods-09-00694-t003:** Change in digestibility of the extracted onion quercetin and grape resveratrol by different methods during in vitro human digestion.

	Small Intestine	
	Quercetin (%)	Resveratrol (%)
Homogenized	45.68 ± 6.45	52.45 ± 3.48
Water extract	40.14 ± 4.65	51.08 ± 5.54
Ethanol extract	46.48 ± 6.04	49.54 ± 3.47

Data are represented as the mean ± standard deviation. *n* = 3.

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
