# Peer review of "Changes in the Content and Bioavailability of Onion Quercetin and Grape Resveratrol During In Vitro Human Digestion"

_foods, 2020, doi:10.3390/foods9060694_

Round 1
Reviewer 1 Report
In my opinion, the major weakness of the manuscript entitled “Changes in the Content and Bioavailability of Onion Quercetin and Grape Resveratrol During In Vitro Human Digestion” is that it addresses an issue on which there are many published studies and consequently, it is difficult to be original/fresh. The bioavailability of onion quercetin and grape resveratrol is well known. However, I consider that it could be interesting to make a comparison between the results obtained with the in vitro system proposed by authors and those obtained with in vivo systems. Or alternatively, authors could explain that their main objective was to compare the bioavailability of onion quercetin and grape resveratrol according to the extracts preparation methods and consequently, focus the discussion in that regard. Taking into consideration these reasons, I consider that the manuscript is not suitable for publication in Foods in the present form.
Other considerations are:
- In section 2.3 In vitro digestion system (specifically on lines 75-76) it is not clear how the membrane dialysis was used (what is put inside and what is outside). This information is successively clarified on lines 87-89. I recommend merging subheadings 2.3 and 2.4.
- In Table 1, it would be better to indicate the desired concentration of each enzyme (α-amylase, pepsin, pancreatin, lipase) and not the added quantity. The amount to be used may depend on the activity of the enzyme.
- Line 95: please clarify that determinations were carried out in all the fractions obtained after mouth, stomach and intestine digestion.
- Lines 90-91 lines: authors state that "The digestibility percentage was calculated as: (sample concentration inside the dialysis tube) / (sample concentration inside the dialysis tube) × 100".
In my opinion the correct way to calculate the digestibility percentage would be: sample concentration outside the dialysis tube / (sample concentration outside the dialysis tube + sample concentration inside the dialysis tube). Please check.
- Lines 127-128: the phrase “The amount 127 of quercetin decreased after small intestine digestion, however, the quantification of quercetin did 128 not change during in vitro human digestion, with the main peak of m / z 64.1 [M] + on LC / MS spectra ”, is not clear.
Author Response
Please see the attachment.
Q1: In section 2.3 In vitro digestion system (specifically on lines 75-76) it is not clear how the membrane dialysis was used (what is put inside and what is outside). This information is successively clarified on lines 87-89. I recommend merging subheadings 2.3 and 2.4.
A1: Thank you for your review and we agree with your comment.
We revised the section 2.3. contents and other subheading according to your comment.
→ p.3, line 74-85 “The in vitro human digestion system was constructed as described previously method… The digestibility percentage was calculated as: {sample concentration inside the dialysis tube/( sample concentration outside the dialysis tube + sample concentration inside the dialysis tube)} × 100.”
→ p.3, line 73, “2.3. In vitro digestion system and digestibility in small intestine”
→ p.4, line 91, “2.4. Measurement of quercetin and resveratrol by high-performance liquid chromatography (HPLC)”
→ p.4, line 104, “2.5. Liquid chromatography-mass spectroscopy (LC-MS) analysis”
→ p.4, line 111, “2.6. DPPH radical scavenging activity”
→ p.4, line 118, “2.7. Statistical analysis”
Q2: In Table 1, it would be better to indicate the desired concentration of each enzyme (α-amylase, pepsin, pancreatin, lipase) and not the added quantity. The amount to be used may depend on the activity of the enzyme.
A2: Thank you for your review and we agree with your comment. We indicated the activities of digestive enzymes (α-amylase, pepsin, pancreatin, lipase) according to your comment.
→ p.2, line 58-59, “α-amylase (~50U/mg), pepsin(≥ 250 units/mg), pancreatin (4× USP/g), lipase (100-500 units/mg),”
Q3: Line 95: please clarify that determinations were carried out in all the fractions obtained after mouth, stomach and intestine digestion.
A3: Thank you for your review and we agree with your comment. We more clearly wrote the method according to your comment.
→ p.4, line 101-103, “The above mentioned each sample means compartment sample resulting from after in vitro digestion step (mouth, stomach and small intestine) with quercetin in onion and resveratrol in grape.”
Q4: Lines 90-91 lines: authors state that "The digestibility percentage was calculated as: (sample concentration inside the dialysis tube) / (sample concentration inside the dialysis tube) × 100".
In my opinion the correct way to calculate the digestibility percentage would be: sample concentration outside the dialysis tube / (sample concentration outside the dialysis tube + sample concentration inside the dialysis tube). Please check.
A4: Thank you for your review and we agree with your comment. We revised the digestibility formula.
→ p.3, line 84-85, “{sample concentration inside the dialysis tube/( sample concentration outside the dialysis tube + sample concentration inside the dialysis tube)} × 100.”
Q5: Lines 127-128: the phrase “The amount 127 of quercetin decreased after small intestine digestion, however, the quantification of quercetin did 128 not change during in vitro human digestion, with the main peak of m / z 64.1 [M] + on LC / MS spectra ”, is not clear.
A5: Thank you for your review and we agree with your comment. We clearly revised the phrase.
→ p.5, line 128-131, “The amount of quercetin decreased after small intestine digestion, however, the mass spectrum of quercetin after each in vitro human digestion steps did not change based on a quercetin main peak with m/z 64.1 [M]+ on LC/MS spectra.”
→ p.5, line 138-139, “however, the mass spectrum of resveratrol decreased after small intestinal digestion based on a resveratrol main peak with m/z 102.2 [M]+ on LC/MS spectra.”

Reviewer 2 Report
Sample of onion: Was the used onion peeled or it was used with dry skin for sampling? Do you analyse only free quercetin? According your results the content of quercetin in onion is quite high, but I guess free quercetin is not predominant form of quercetin in fresh onion, there are the its glycosides. Is it possible that this high content would be generated during sample processing (drying for two days under moderate temperature)?
Analytical method: The chromatographic conditions given in Chapter 2.5. do not correspond the conditions given in Figures 1 and 2.
Author Response
Please see the attachment.
Q1: Sample of onion: Was the used onion peeled or it was used with dry skin for sampling? Do you analyse only free quercetin? According your results the content of quercetin in onion is quite high, but I guess free quercetin is not predominant form of quercetin in fresh onion, there are the its glycosides. Is it possible that this high content would be generated during sample processing (drying for two days under moderate temperature)?
A1: Thank you for your review and we agree with your comment.
We used onion and grapes including skin, and all samples were dired at 45°C for 48h.
Actually, we analyzed total quercetin contents without distinction free quercetin and quercetin glycoside. Previous study reported that quercetin and quercetin 3-glucoside in commercial chartreuse onion were 127.92 and 24.16 mg/g, respectively (1). In addition, other study indicated that thermal processing or heat treatment were observed increased amount of the free flavonoids because the processing might produce alteration in their extractability by the disruption of the plant cell wall (2). Thus, the high content quercetin could be considered by more easily releasing depends on sample processing.
Thus, sample precessing might be contribute to alteration in quercetin content.
→ p.8, line 179-182, “In addition, previous study also reported that quercetin and quercetin 3-glucoside in commercial chartreuse onion were 127.92 and 24.16 mg/g, respectively [27]. However, this study was analyzed total quercetin contents without distinction free quercetin and quercetin glycoside.”
Related paper
(1) Kwak, J. H., Seo, J. M., Kim, N. H., Arasu, M. V., Kim, S., Yoon, M. K., & Kim, S. J. (2017). Variation of quercetin glycoside derivatives in three onion (Allium cepa L.) varieties. Saudi Journal of Biological Sciences, 24(6), 1387-1391.
(2) Choi, Y., Lee, S. M., Chun, J., Lee, H. B., & Lee, J. (2006). Influence of heat treatment on the antioxidant activities and polyphenolic compounds of Shiitake (Lentinus edodes) mushroom. Food chemistry, 99(2), 381-387.
Q2: Analytical method: The chromatographic conditions given in Chapter 2.5. do not correspond the conditions given in Figures 1 and 2.
A2: Thank you for your review and we agree with your comment. We revised the wrong method and figure legends according to your comment.
→ p.4, line 93-95, “a Fortis H2O column (250 mm × 4.6 mm, 3 μm) with a linear gradient of solution A (acetonitrile, 0-70%) and solution B (water containing 0.1% formic acid) at a flow rate of 1.5 mL/min were determined.”
→ p.4, line 97-99, “For resveratrol, the mobile phase was determined with a linear gradient of solution A (acetonitrile, 0–70%) and solution B (water containing 0.2% formic acid). The UV detector wavelength was set at 306 nm, the flow rate was 0.8 mL/min, and the injection volume was 20 μL.”
→ p.6, line 152-153, “All HPLC operation was carried out with a linear acetonitrile gradient (0–70%) and 0.1% formic acid in water, and at a flow rate of 1.5 ml/min using a UV detector at 370 nm.”
→ p.7, line 158-160, “All HPLC operation was carried out with a linear acetonitrile gradient (0–70%) and 0.2% formic acid in water, and at a flow rate of 0.8 ml/min using a UV detector at 306 nm.”

Round 2
Reviewer 1 Report
The authors improved the manuscript taking into account most of the reviewer's comments.
Reviewer 2 Report
I have no further comments.